# 100 Years of Chromosome Research in Rye, *Secale* L.

**DOI:** 10.3390/plants11131753

**Published:** 2022-06-30

**Authors:** Rolf Schlegel

**Affiliations:** Julius Kühn Institute, Federal Research Centre for Cultivated Plants, Erwin-Baur-Str. 27, 06484 Quedlinburg, Germany; rolf.schlegel@t-online.de

**Keywords:** *Secale*, rye, chromosome, karyotype, meiosis, tetraploidy, translocation, trisomics, GISH, CRISPR/Cas

## Abstract

Although microscopy and genetics were still in their infancy, there are cytological results produced a hundred years ago that are still relevant today. Since the 1920s, rye has been a subject of chromosome research. It started by plotting its mitotic and meiotic chromosomes to determine genome size. After controversial evidence, it became clear that the base number is *n* = 7. However, structural differences exist between species within the genus *Secale*. Some rye populations even carry accessory chromosomes evolutionary derived from the A genome. The development of tetraploid strains significantly promoted chromosome analysis. Various techniques have tried to stabilize the disturbed chromosome pairing of the induced tetraploids. Although slight improvements could be achieved, they did not lead to a breakthrough. However, the various aneuploid derivatives of the polyploids found major advances in the genetic analysis of rye. Trisomics, telo-trisomics, and reciprocal translocation have served as important tools for gene mapping. Since the 1970s, various chromosome banding techniques have stimulated scientific progress. The seven haploid chromosomes could be diagnosed unequivocally, not only in *S. cereale* but also in related species. These findings led to a clear homoeologous assignment to the genomes of related grass species such as wheat, barley, rice, etc. Current applications of in situ fluorescence staining methods, such as GISH and FISH, allow even more precise results, depending on the specificity of the DNA samples. Advanced preparation techniques are supplemented by the variety of innovations in the field of molecular genome analysis. They replace complex cytological examinations. In this way, introgressions can be safely detected by DNA markers and be much more detailed. In addition, CRISPR/CAS-mediated chromosome engineering will become an important method of the future.

## 1. Introduction

Rye is one of the Old World grains that was the last to reach Europe, but later, it became the most important cereal for bread-making of the Middle Ages. Historically, rye has not only significantly influenced the development of Western European culture, but it was also one of the first objects of modern cell research in plants. Its status as an important crop, its diploid chromosome set, low chromosome number, and the size of its chromosomes made it so. Two monographs on this topic have been published in the past few years, which already contain many of the latest developments in rye [1,2]. Therefore, the current compilation is limited to the presentation of the methods of chromosomal analysis with regard to genetic and breeding applications.

During the 1920s, the first pictures of rye chromosomes were published. It is therefore a good opportunity to take stock after 100 years of research.

## 2. Karyology

Although initially, haploid chromosome numbers of *n* = 8 were counted, it was soon found that there are only *n* = 7. Nemec [3] studied triploid endosperm cells of rye and found 18 chromosomes. Therefore, he concluded that the haploid number should be *n* = 6, i.e., 3 × 6 = 18 chromosomes. He also recognized early that this chromosome number is inherited quite consistently, despite spontaneous aberrations that are occasionally observed. Nakao [4] drew the first meiotic prophase chromosome of rye after Heidenhaill’s iron alum hematoxylin staining (Figure 1). Even hybrids between wheat and rye he already studied by this technique. Usually, he counted eight pairs of chromosomes. As compared to other plants, “*in rye the diakinetic figure is very curious. Each double chromosome often connects with others forming irregular groups or rings*.” … “*In rye many small nucleoli, besides one large one are present, and attached mostly to chromosomes*”, Nakao wrote.

However, regarding the chromosome number of rye, he was wrong. Sakamura [5,6] corrected the counting from eight to seven haploid chromosomes, which was confirmed by Kihara [7] (Figure 2). The latter author already observed irregular meiotic chromosome pairing in wheat–rye hybrids as reason for partial sterility.

In 1924, Gotoh [8] tried again to clarify the occurrence of rye forms with seven and eight chromosomes. Two homologous chromosomes of the eight-chromosome plants very often behave differently from the 14 others in the pollen mother cells. He came to the conclusion that they are a result of transverse division of two specific chromosomes of the seven-chromosome complement (Figure 3). Gotoh could not know yet that there are so-called A chromosomes and so-called B chromosomes in rye, which differ both in shape and meiotic behavior. In the early rye populations, especially East Asian populations, many were later described as carriers of B chromosomes [9]. Thus, Gotoh [8] was almost right in his description of the accessory chromosomes.

After it was clarified in the mid-1920s that the haploid base number in rye is *n* = 7, attempts were made to study the chromosomes between the morphologically classified rye species. Previously, Lewitzky [10] described in detail the morphology of the somatic chromosomes. He found metacentric and sub-metacentric chromosomes as well as a so-called satellited chromosome. He was first numbering the rye chromosomes from I to VII, wherein chromosome II was the satellited one with a secondary constriction (Figure 4).

### Wild Species

E. K. Emme [9] started to compare the related species of rye that were collected before by the Russian botanist N. I. Vavilov and co-workers [11,12,13] in 1915 and 1920. They included in their studies common rye, *Secale cereale*; mountain rye, *S. montanum*; forest rye, *S. silvestre*; and African rye, *S. africanum*. The measurements of mean chromosome length revealed differences between the species. Besides *S. cereale, S. montanum* seemed to carry the largest chromosomes. The samples were classified as rye with brittle spikes and non-brittle spikes. Since *S. cereale var. afghanicum* showed a somewhat longer chromosome pair, the author concluded that the extra segment carries the gene for brittle spikes. It could be the first approach of physical mapping of rye genes! Moreover, the author described two satellite chromosomes in *S. montanum* as well as so-called strip-like bivalents, which obviously were multivalents (Figure 5).

Since the beginning of the 20th century, botanists have tried to morphologically classify the different forms of rye, which has led to much controversy. It took more than 50 years until the different species were deliberately crossed with each other. It was found that the species also differed chromosomally. Not only the size and the centromere position were different, but also the structure. Multivalents were observed in the interspecific hybrids during meiotic chromosome pairing. This pointed to so-called translocations of chromosome segments. Kostoff [14] was one of the first to report interspecific crosses. His results were later confirmed by Nakajima [15].

Meanwhile, it is known that the various types of rye can be differentiated by their DNA content, base sequence, and chromosome structure. The complex species relationships are shown in Figure 6.

## 3. Polyploidy

With the discovery of colchicine as a mitotic poison [17,18], botanists and plant breeders tried to improve cultivated plants by genome duplication. It was already known that this leads to gigantism, i.e., larger cell size, as well as larger plants as a whole. A spontaneous triploid rye was already reported by Nemec [3]. Chin [19] and Dorsey [20] first reported autotetraploid rye induced by colchicine. This was followed by a large number of reports [21].

However, the hope of the breeders was not fulfilled. Although the tetraploids consistently showed stronger vegetative growth, the fertility of the plants fell. A variety of chromosomal disorders occurred during meiosis, which reduced the vitality of pollen and oocytes. Besides bivalents, multivalents and chromatid bridges occurred, and the chiasma frequency was reduced, resulting in aneuploid embryo formation. Aneuploids again showed jagged spikes with low seed setting and yield. In order to avert this handicap, extensive research of chromosome pairing was established.

### 3.1. Genetic Control of Chromosome Behavior

While genes as permanent hereditary units of the nuclear genotype do not heavily change, the chromosomes that carry them often change their behavior and shape. In this respect, the genetic complement presents a phenotypic character. It changes both during cell division and throughout the life cycle. Thus, there is a permanent interaction between genotype and the environment.

In rye, meiosis can be less efficient in inbred rye than in population rye. It can differ with respect to frequency, terminalization, and localization of chiasmata (Figure 7). There are changes in neocentric activity as well as premeiotic errors. Chromosome breakages occur at first prophase. Chiasma frequency and terminalization are polygenically controlled and correlated with chromosome pairing [22,23,24,25]. This proved early that cytogenetic research is important for rye breeding, i.e., regular chiasma formation not only guarantees normal chromosome distribution during meiosis but also secures genetic recombination and finally yield.

The formation of chiasmata differs between the chromosome arms. The highest frequencies can be found in 1RS, 1RL, 3RS, 3RL, 7RS, and 7RL, while the arms 2RS, 4RS, 4RL, 5RS, and 6RS significantly form less chiasmata. Moreover, the chiasma frequency is generally reduced (about 10%) in proximal regions as compared to telomeric regions, as described by Naranjo [26]. The latest analysis with tetrad fluorescence in situ hybridization combined with linkage maps [27] indicated that heterochromatin strongly suppresses recombination. The mean chiasma frequency in pollen mother cells (PMC) and egg mother cells (EMC) was almost identical, with about 9 Xta/PMC [28].

The prerequisite for the formation of the chiasmata is an exact homologous pairing, i.e., synapsis of the appropriate chromosomal regions. It is coupled to a mechanism that repairs chromosomal breaks during the crossing-over process. In rye, the synapsis begins near the ends of the chromosomes and also leads to predominantly terminal chiasmata.

Both the mitotic and meiotic chromosomes are characterized by a basic structure of “parallel fibers” and “chromomeres”, which are highly compact during the metaphase. The meiotic chromosomes are much more condensed and have a somewhat smoother surface [29].

It was recognized early that the chromosomes in the cell nucleus are not randomly arranged but follow a specific spatial order. For example, the telomeres show a bouquet-like distribution that corresponds to the so-called Rabl configuration [30,31,32]. Overall, several premeiotic phases can be described. In the leptotene, the chromosomes are still found as long threads in the nucleus, with sister chromatids lying close together. Between them, there is a proteinaceous axial structure, which apparently provides cohesion.

Genetic recombination begins during the zygotene. For a short period, the sister chromatids become visible after the chromosomes are condensed by subsequent coiling. The regulation of this pairing phase seems to be influenced by specific proteins of the chromatin, e.g., meiotin I. A ribbon-like structure is now formed between the homologous partners over the entire length of each pair of chromosomes, which is referred to as a synaptonemal complex. The latter consists of axial, lateral, and transverse elements that are connected to each other. Although the rye chromosomes carry large heterochromatic regions at the ends, the synaptonemal complex formation follows the entire length [33]. The large telomeres seem to not interfere with the synaptic process.

In the later zygotene, the synaptonemal complex is completed from one chromosome end to the other. The length of the axial structure agrees well with the respective arm lengths of the chromosomes, which can be measured at the bivalents in the pachytene. On average, there are more than ten initiation sites along a bivalent, which serve to complete the pairing of the two chromosomes. The individual bivalents are still spatially ordered within the cell nucleus.

During the early pachytene, the close homologue association becomes weaker. Then, the so-called recombination nodes become visible, which can be larger, elliptical, or smaller. Their number and position along the bivalents correlate with the later number and distribution of the chiasmata [34].

Advances in microscopic techniques, meanwhile, allow deeper insights into the mating process during meiosis. The structural modifications during the prophase can be observed using electron and super-resolution microscopy combined with immunohistochemistry and FISH [35]. Four proteins are of particular importance during prophase pairing (HEI10, NSE4A, ZYP1, ASY1). They contribute to the synapsis and formation of the synaptonemal complex. They are involved in the degradation and dissolution of the lateral and axial elements of the central region of the synaptonemal complex, and they participate on desynaptic events, chromosome condensation, proper recombination, as well as separation of homologous chromosomes. The investigations also showed that fragmentation of the synaptonemal complex and formation of the ball-like structures (recombination nodes) occur during late diakinesis. Hesse et al. [36] demonstrated that the mating process during prophase and the formation of the synaptonemal complex in the B chromosomes of rye are similar to the A chromosomes. Based on their research, they proposed an extended mating model (Figure 8).

### 3.2. Manipulation of Chiasma Frequency

When it became clear that complete chromosome pairing in autotetraploid rye depends on the number and localization of the chiasmata, attempts were first made to increase the chiasma frequency. However, several authors observed chromosome associations in diploid and tetraploid rye that are not correlated with the frequency of chiasmata [36,37,38,39]. This is true both for pollen mother cells (PMCs) and egg mother cells (EMCs). Their frequencies with about 9 Xta/PMC are almost identical. It was suggested that chiasma formation in PMCs and EMCs is governed and regulated by a single controlling system of genes, and that variation in this genetic system is expressed identically in the two sexes [28]. Only Hazarika and Rees [40] came to the conclusion that the frequency of chiasmata per chromosome determines the type and number of multivalent configurations in tetraploid rye (Figure 9), which could not be confirmed by numerous experiments.

During the early 1970s, efforts were made to exploit the potential of autotetraploid rye for breeding purposes. Instead of inducing autotetraploids with colchicine, they should be provided in large numbers by valence crossings. Here, tetraploid plants as the maternal partner were crossed with diploid plants as the paternal partner. In the F1 offspring, in addition to triploid plants, some tetraploid plants can also be selected, which result from fertilization with an unreduced gamete (pollen grain) [41]. In this way, it was possible to select diploid parent plants that, e.g., were characterized by particularly high chiasma frequencies per pollen mother cell. The evaluation of numerous experiments showed that the chiasma frequency/PMC of the diploid mating partner had only a minor influence on the formation of complete quadrivalents during meiosis. The fertility of the induced tetraploids and the frequency of aneuploids were not affected, either.

Nevertheless, significant differences in the frequency of multivalents/PMC could be observed in the offspring. A more detailed analysis showed that the chromosome structure of the diploid crossing partners played a role. The diploid crossing partners came from different, genetically distant rye populations. The karyological differences that could be determined with the microscope were small. However, it appeared that there was a preferential pairing of chromosomes within the parental genomes. Thus, the tetraploid F1 plant showed more bivalents/PMC (Figure 10). These findings supported the theory of Sybenga [42]. He suggested that preferential diploid-like bivalent pairing could be increased by structural changes via induced translocations.

### 3.3. Manipulation of Chromosome Structure

Sybenga produced a series of reciprocal interchanges between rye chromosomes. He applied X-ray irradiation in order to induce the chromosome breakages in pollen grains. The latter were used for pollination of plants of inbred lines derived from his standard genotype Petkus. In F1 progeny, the quadrivalent configurations can be identified resulting from the reciprocal exchange of nonhomologous chromosome segments (Figure 11). The verification of translocated chromosomes was facilitated by test crossings with disomic wheat–rye chromosome addition lines of Chinese Spring-Imperial. Interchanges were often described in rye, but found in genetically different populations [43]. They were less suitable for genetic and cytological studies.

Over many years, Sybenga and his co-worker created a complete set of translocation lines, involving all seven rye chromosomes, as shown in Table 1 [44]. This tester set served not only for analysis of genes and their localization [14], but also for experiments on induced diploid-like chromosome pairing in autotetraploid rye. Several defined translocations were combined homozygously into single inbred lines. This resulted in a complex structural heterozygosity that partially led to preferential bivalent formation but also to a severe drop in fertility of the tetraploid plants. Other efforts also failed. Augustin and Schlegel [45] also created more than 20 homozygous lines with reciprocal translocations of the Petka cultivar. However, they tried to select lines whose exchanged chromosome segments were as small as possible. Although this made cytological identification more difficult, it caused fewer disturbances during the meiotic chromosome pairing. Apart from that, it was impossible to handle such experimental lines in breeding. The cytological effort would have been enormous. Thus, improving the fertility of tetraploid rye was a long way off.

Pericentric inversions are another type of genome rearrangement. They were detected in chromosome 3R and 4R of rye [46] and recognized as the most rapidly evolving chromosomal regions. The play a major role in the genome evolution of Triticeae.

### 3.4. Trisomics and Telo-Trisomics

If different ploidy levels are crossed with each other, then triploids arise among other chromosome numbers. Their offspring are highly aneuploid. Plants with one additional chromosome are most viable and frequent. They are designated as primary trisomics [47]. They can also arise spontaneously from chromosome mismatches during meiosis of diploid plants. They were described as early as 1935 [42,48]. Kamanoi and Jenkins (1962) first described a complete set of primary trisomics [49], but it was lost later on. From the winter rye varieties Danae and Heines Hellkorn, two new series of trisomics were produced [50,51,52]. Both series were derived from backcrossing of triploid plants to diploid plants. By microscope, the extra chromosomes can be counted as and identified (Figure 12). In addition, each additional chromosome causes more or less pronounced morphological effects. This allows the estimation of quantitative allelic effects of individual chromosomes and sometimes the morphological identification of the trisomic line.

After self-pollination of trisomic plants, a tetrasomic offspring cannot be expected. The strong manifestation of the incompatibility of the cross-pollinator rye is one reason. The second reason is that extra chromosomes are rarely tolerated by the pollen. Only the egg cells allow transmission in about 15% of cases. Thus, trisomics can only be propagated via permanent backcrossing with diploid plants. In addition to other aberrations, monotelocentric chromosomes can be found during backcrossing. They arise from a maldistribution of the trisomic, regardless of whether it was paired or univalent. The trisomic chromosome often pairs with its two homologous partners, usually as a chain trivalent. Thus, in addition to the trisomics, monotelocentric series can also be established. Almost all of the 14 possible chromosome arms have been described [53,54,55,56].

Before molecular methods for gene mapping became important in the 1990s, the primary trisomics and telotrisomics were the most important tools of gene analysis in rye [57,58,59].

## 4. Chromosome Staining

### 4.1. Radioactive Labeling

In order to demonstrate semi-conservative replication and uninemy of chromosomal [47] DNA in rye, autoradiographic studies using tritium-labeled thymidine can be applied. Cells are thymidine-labeled during the synthesis phase, incorporated into the newly synthesized DNA upon replication. After two cell cycles, the metaphase chromosomes then each consist of a marked one and an unlabeled chromatid, which was first autoradiographed by Lima de Faria [60]. The author compared common chromosome spreads after Feulgen staining with thymidine-labeled chromosomes after growing rye seedlings in a solution of tritiated thymidine. Leaf nuclei showed heavy heterochromatic regions in proximal positions, while euchromatin was observed mostly distal. The DNA synthesis of euchromatin and heterochromatin takes place asynchronously, both within and between the chromosomes. When the silver grains were evaluated from the autoradiographs, a two to three times bigger number was noted in the euchromatin as compared to the heterochromatin, demonstrating a higher synthetic activity. On the other hand, the heterochromatin had two to three times more DNA than the euchromatin when the findings from the Feulgen preparations were compared with those from the radioactively labeled ones. However, this method required a relatively large amount of technical effort and only showed a low resolution due to the scattered radiation.

### 4.2. C/N-Banding

The first banding studies were published by Sarma and Natarajan [61]. They applied both C-banding and fluorescence banding. It became obvious that the rye chromosomes show very little heterochromatin in the centromeric regions compared to wheat, barley, or oats. This and the prominent telomeric C-bands in rye allow an easy differentiation of the chromosomes in hybrids between wheat and rye, e.g., in triticale.

A short time later, Gill and Kimber [62] demonstrated that the distribution of euchromatin and heterochromatin within the seven chromosomes has diagnostic values (Figure 13). The distribution harmonized well with the detailed data of chromomere distribution [63]. The application of chromosome C-banding for meiotic chromosomes and their pairing behavior was first described by Schlegel and Friedrich [64].

Although the first results on Giemsa C-banding allowed a more detailed description of rye chromosomes, there was still confusion about their designation. The high degree of structural polymorphism, promoted by the allogamous mode of pollination [65], increased the uncertainty.

Nevertheless, the banding studies, together with conventional comparisons and the introduction of a standard karyotype, contributed to an universal chromosome nomenclature in rye. As shown in Table 2, most of the karyological results from the past could be brought into broad agreement.

Still missing was the approval of the people who were dealing with rye chromosomes at the time. After various preliminary talks, Prof. Jaap Sybenga from Wageningen (The Netherlands) agreed in 1983 to organize a workshop on “Rye Chromosome Nomenclature and Homoeology Relationships”. The participants of this workshop from Europe, America, and Australia agreed to (a) designate the seven different rye chromosomes from 1R to 7R, (b) to consider the homology of the rye chromosomes within the Triticeae, and (c) to accept the seven wheat–rye chromosome addition lines of the series Chinese Spring-Imperial as the standard [75].

In addition to these agreements, it was proposed to establish a standard genotype from rye itself, what seemed difficult because of the strong self-incompatibility in rye. The author was entrusted with this task. The wheat–rye additions were not suitable as a standard for all cytogenetic approaches. They also accumulated chromosomal changes in the course of their maintenance.

Based on the globally known and widely used Petkus gene pool, haploid, dihaploid, and tetrahaploid standard rye from the Petka variety were produced [71]. The spring-type Petka was released in 1961, showing predominantly greenish grains, and the habit corresponds to that of a modern variety. By diploidizing the haploid and/or dihaploid plants, it was possible to produce genetically and structurally homozygous genotypes.

The measurement of the haploid chromosomes resulted in a pattern as shown in Figure 14. The longest chromosome is 2R and the shortest is 1R. The arm indices vary between 1.0 (3R) and 2.1 (5R); thus, centromeres are inserted proximal (3R) to subterminal (5R, 6R). The sizes of centromeres are related to the genome size [76].

Together with significant morphological differences, all seven chromosomes show characteristic C-band patterns, in addition to typical N-bands on chromosomes 2R, 3R, and 6R [77]. Including a chromosome band nomenclature [78] and excluding structural heterozygosity [18], a reference karyogram was established.

During the 1990s, a large number of molecular studies on cereals began. It soon became apparent that the arrangement of genes in different species is quite similar and has been conserved over a long period of evolutionary time. In the case of wheat, barley, rye, millet, or rice, this gene order is only broken by larger segment exchanges within the genome. Some compact noncoding DNA sequences evolved that are genome-specific. They are frequently concentrated at the chromosome ends, e.g., in rye. The latter shows more than 10 larger interchanges, as can be seen in Figure 15. They obviously arose during its evolution, which lasted about 6 million years [79].

### 4.3. Sister Chromatid Exchange

The incorporation of 5-bromodeoxyuridine (BrdU) into chromosomal DNA leads to a reduction in UV fluorescence after staining with a fluorescent dye, e.g., “H33258”. If cells are cultivated for two cell cycles in a medium containing BrdU, this is incorporated into the DNA instead of thymidine, so the metaphase chromosomes then consist of a chromatid in which the thymidine is in only one DNA strand and the chromatid in the other is substituted in both DNA strands by BrdU.

After subsequent staining with “H33258”, the doubly substituted chromatid can be distinguished from the monosubstituted chromatid due to the weaker UV fluorescence (Figure 16). Friebe [80] observed a distribution pattern of SCEs fitted with a Poisson distribution. Within telomeric heterochromatin of the satellite chromosome, the observed frequency of SCEs was lower when compared with euchromatic regions, whereas segments adjacent to the heterochromatin show an increased frequency of SCEs. A corresponding distribution pattern could be observed by Schlegel and Friedrich [64] and Jones [81] for the meiotic exchange, where the chiasmata could be localized preferably between the terminal heterochromatin blocks and the neighboring euchromatic chromosome sections, but very rarely in the area of the heterochromatin.

### 4.4. Non-Radioactive Labeling

The application of florescence in situ hybridization (FISH) has significantly expanded the chromosome studies. At the beginning, total genomic DNA was used as probes (GISH). If the genetic differentiation between the species is large enough, then the parental genomes in hybrids between species and genera can be well distinguished within one somatic or meiotic cell. A first attempt was made by Schwarzacher et al. [82] when they examined rye from barley chromosomes in a hybrid of *Hordeum chilense* x *Secale africanum*. GISH experiments soon followed by Bashir et al. [83] in wheat–rye addition line 7R, Heslop-Harrison et al. [84] in the 1BL.1RS wheat–rye translocation, and Schlegel et al. [85] in the 4BS.4BL-5RL wheat–rye translocation. The latter even enabled the first physical mapping studies in rye (Figure 17). A short terminal segment of chromosome arm 5RL was interchanged with the long arm of chromosome 4B of Viking wheat carrying several marker genes. Until now, numerous projects followed this approach [86].

Introgression of rye chromosomes into different types of wheat is still an important field of breeding and research [88]. However, rye itself also became a subject of genetic introgressions. A set of rye–wheat chromosome addition lines was produced [89,90]. Even a stable translocation between chromosomes 6B and 2R could be experimentally created in rye by Schlegel and Kynast [91] (Figure 18). However, the effect of wheat genes on chromosome pairing and breeding characteristics was limited [92,93,94].

When different probes are used, e.g., pSc200, pSc250, CCS1, 5S rDNA, or 25S rDNA, as described by Mikhailowa et al. [85], defined positions of centromeres, subtelomeric domains, and rDNA sites on chromosomes can be defined. The latest non-denaturing fluorescence in situ hybridization technology provides a convenient and efficient way to identify individual rye chromosomes. However, the number of suitable FISH-positive oligo probes for recognizing specific segments of rye chromosomes is still limited. Xi et al. [95] demonstrated new probes that even revealed tandem repeats in rye highly diagnostic not only for chromosomes but also chromosome arms, e.g., the short arms of 1R, 5R, and 6R and the long arms of 4R and 7R.

Additional probes, together with multicolor staining and high-resolution microscopy, provided new insights on chromosome structure and its evolution [1]. This was impressively confirmed when studying B chromosomes. Although their DNA content contributes about 5.4% to the basic genome [96], it seems that they are transcriptionally inactive. This can be explained by a loss of coding sequences. Selective amplification of repetitive DNA might be another reason.

B chromosome-specific DNA could be revealed by amplified fragment length polymorphism (AFLP) studies. Its share is about 1% of the total DNA, much less than expected. The characterized fragments are highly repetitive. When they were used in FISH analysis, they could be found throughout A and B chromosomes. Some of dispersed fragments can be detected both on A and B chromosomes with almost equal density. Another type of dispersed fragments was found to be concentrated near the centromere on the long arm of the standard B chromosome, which implies that this region is less complex. It could be an amplification hot spot of specific sequences.

The heavy block of terminal heterochromatin of the B chromosome contains both specific and non-specific B chromosome DNA [97,98]. One of the repeat samples that is B-specific was located within terminal heterochromatin (Figure 19). However, most DNA that appears B-specific was observed at the subtelomeric region identical to a heterochromatin band diagnosable at metaphase I [99,100].

Although the B chromosomes emerged from the A genome, they developed independence in the course of evolution. This does not exclude the possibility that they can again re-associate with the A chromosome. Schlegel and Pohler [101,102] demonstrated a spontaneous translocation between the standard B chromosome and the chromosome 3RS (Figure 20).

## 5. In Situ Manipulation

In order to acquire specific DNA of the B chromosomes, attempts were first made to obtain it by means of in situ extraction chromosomes. As demonstrated in Figure 21, single chromosomes or chromosome fragments were taken with a micro-pipette from a microscopic chromosome spread [103]. The method requires sophisticated equipment and a skilled taxidermist that few laboratories have. Therefore, alternatives were developed, which were found with the so-called flow sorting (see below). The in situ manipulation was thought as a first step for in vivo chromosome manipulation and chromosome transfer, both intraspecific and interspecific. Although the method offered great opportunities for breeding research, it was not widely used. In the meantime, genetic engineering has developed rapidly, making in vivo chromosome manipulation unnecessary.

## 6. Flow Sorting

Using flow cytometry cells, nuclei, or chromosomes from metaphase I, cells are screened according to their fluorescence or light scatter during flow with high speed in a narrow stream of liquid. Hülgenhof et al. [104] applied it first for rye. By sorting cell nuclei, they determined the DNA content as 2C value with 17.3 picograms (pg) of the variety Petka. Their method was later applied in order to differentiate hexaploid triticale karyotypes with lower or higher heterochromatin contents of the rye chromosomes. Reduced amount of terminal heterochromatin in rye chromosomes should stabilize the homologous chromosome association in the genetic background of wheat, i.e., triticale [30,31].

Bashir et al. [83] determined the nuclear DNA content of the seven rye chromosomes present in wheat–rye addition lines with 0.8 pg (3R) to 1.4 pg (7R), and 7.8 pg per haploid genome. This was in agreement with the 2C value of the diploid genome (see above).

Together with improved cell synchronization and chromosome isolation [105], flow sorting became a veritable tool for karyotyping as well as chromosome genomics in rye (Figure 22) [106,107,108,109,110].

## 7. Outlook

Since the first observations of rye chromosomes using simple light microscopes, research on these chromosomes has undergone tremendous development, influenced both by developments in preparation techniques and by the stimulating effects of breeding. This will also be the case in the future. The latest microscopes with image acquisition and processing systems enable the simultaneous two- or three-dimensional, multicolor visualization of both single-copy and highly repetitive sequences in the rye genome. It will be supplemented by deep learning artificial intelligence, as described by Nagaki et al. [111].

Progress in the karyological analysis of rye chromosomes is increasingly determined by the advancement of molecular methods. The large amount of repetitive sequences of the rye genome will allow further insights into both relationships between rye species and their evolution [112]. Many results with simple sequence repeats (SSRs) and/or microsatellites already prove this. Even individual chromosomes and chromosome arms can be identified with them [113].

ISSR markers, which are obtained from nucleotide sequences between microsatellite priming sites, have proven to be another efficient method of genome analysis in rye [1,86,114,115]. Genotyping-by-sequencing coverage analysis is capable of detecting chromosome introgression in detail [116].

Although CRISPR-associated protein (Cas)-mediated gene editing has revolutionized biology and plant breeding, large-scale, heritable restructuring of plant chromosomes is still in its infancy. However, duplications and inversions within a chromosome, and also translocations between chromosomes, can now be achieved by CRISPR/Cas-mediated chromosome engineering. Subsequently, genetic linkages can be broken or can be newly created. Moreover, the order of genes on a chromosome can be changed. While natural chromosomal recombination occurs by homologous recombination during meiosis, CRISPR/Cas-mediated chromosomal rearrangements can best be obtained by harnessing nonhomologous end joining pathways in somatic cells [117].

## Figures and Tables

**Figure 1 plants-11-01753-f001:**
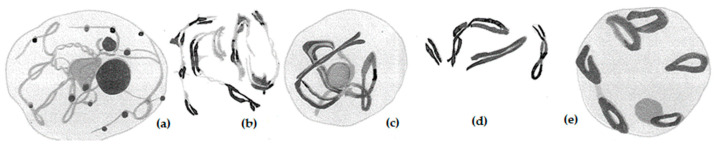
Drawings of meiotic prophase chromosomes of rye, *Secale cereale*, as bivalents with an ABBE drawing device and after fixing with chromic, osmic, and acetic acid, or with acetic alcohol (Flemming’s mixture), and with Heidenhaill’s iron alum hematoxylin staining. (**a**) Small nucleoli during prophase, (**b**,**c**) Late prophase, (**d**) Chromosome formation, (**e**) Diakinesis. Modified after Nakao (1911), with permission.

**Figure 2 plants-11-01753-f002:**
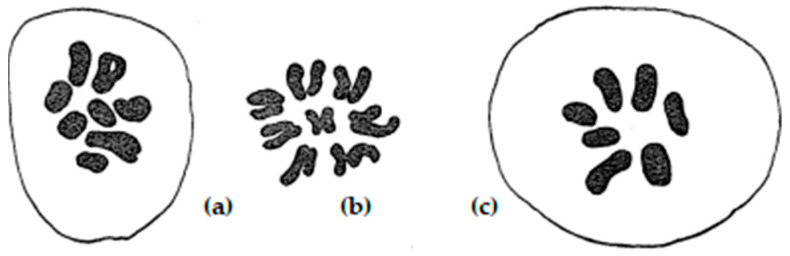
Drawing of meiotic chromosomes of rye, *Secale cereale*, with (**a**) eight bivalents during diakinesis, (**b**) with 8 separating bivalents during early anaphase I, (**c**) with bivalents during metaphase I. Sometimes V-shaped configurations were observed. Modified after Kihara (1919), with permission.

**Figure 3 plants-11-01753-f003:**
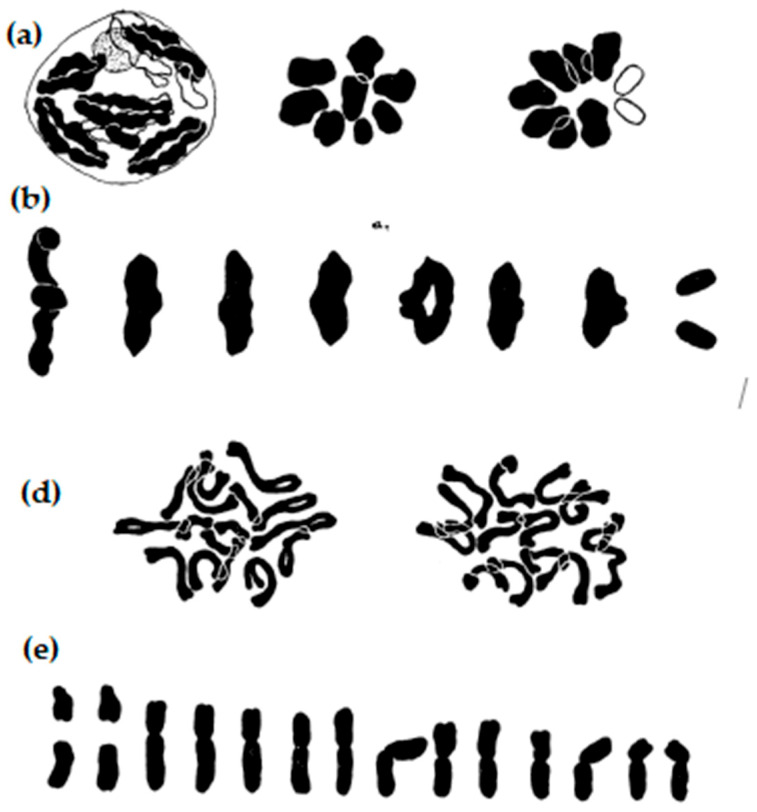
Meiotic and somatic rye chromosomes with eight chromosomes. The somatic spread in the third row, right, shows just 2*n* = 14 chromosomes. (**a**) Bivalents during prophase to metaphase I, (**b**) Individual bivalents as rod and rings plus two univalents, (**d**) Somatic chromosomes from root tips, (**e**) two pairs of smaller somatic chromosomes: one pair with acrocentric chromosomes, one with metacentric. Both are later named standard B chromosomes. Modified after Gotoh (1924), with permission.

**Figure 4 plants-11-01753-f004:**
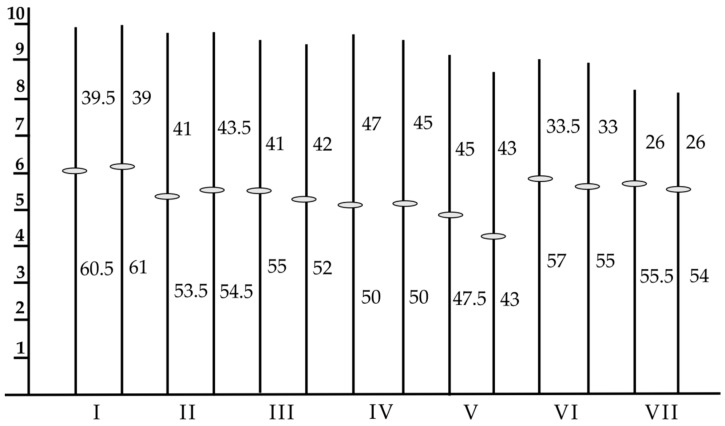
Measuring of 14 somatic chromosomes of rye, *Secale cereale,* var. Vyatka, i.e., chromosomes I to VII. Chromosome II is the satellited one. Modified after Lewitzky (1931), with permission.

**Figure 5 plants-11-01753-f005:**
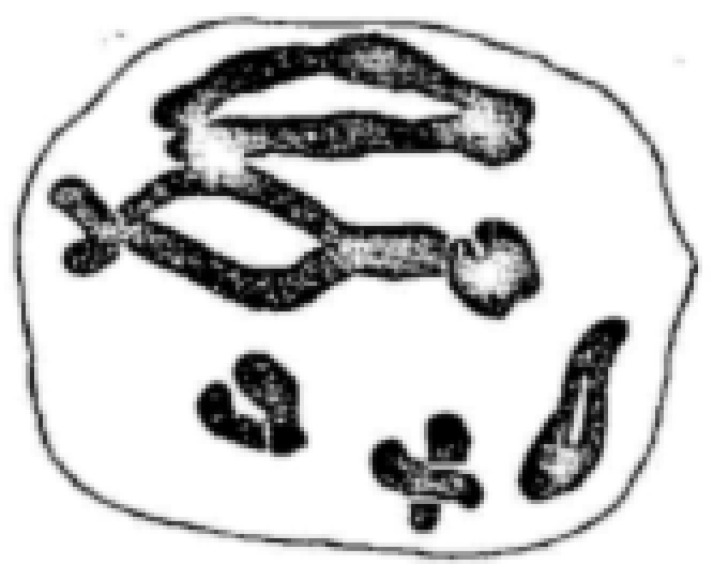
Metaphase I pairing configuration in *Secale montanum* with 4 bivalents, 2 quadrivalents. Modified after Emme (1927), with permission.

**Figure 6 plants-11-01753-f006:**
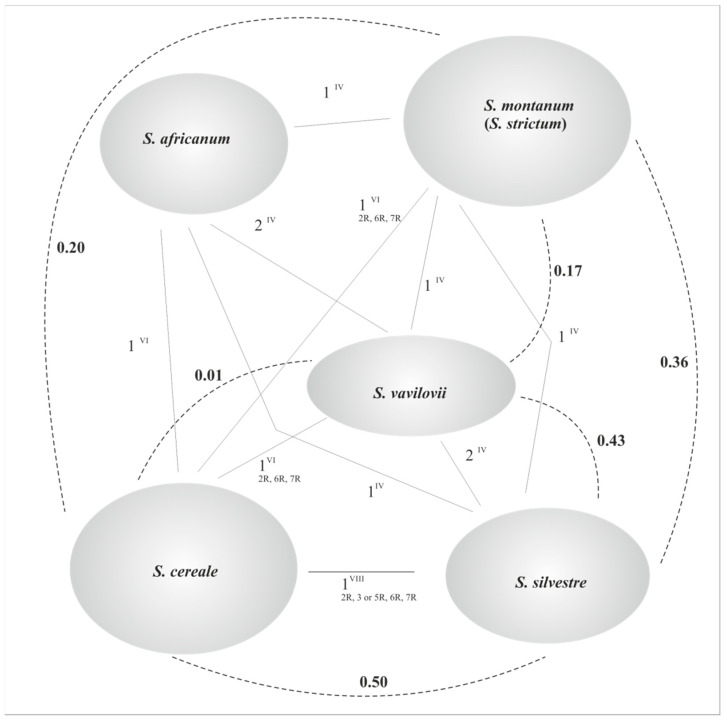
Cytological relationships between species of rye and the chromosomal interchanges between them (IV = quadrivalent, VI = hexavalent, VIII = octavalent). Broken lines show fixation indices (F_ST_) between *Secale* taxa (after Schreiber et al. 2018 [16]); the smaller the value, the closer the genetic relationship.

**Figure 7 plants-11-01753-f007:**
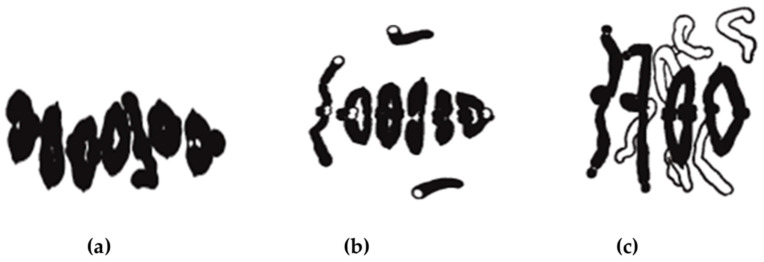
Metaphase I pairing of homologous chromosomes during meiosis of diploid inbred lines of rye with (**a**) 6-ring and 1-rod bivalents, (**b**) 2 univalents and 5-ring and 1-rod bivalents, and (**c**) 6 univalents plus 2-ring and 2-rod bivalents. Modified after Rees (1955a), with permission.

**Figure 8 plants-11-01753-f008:**
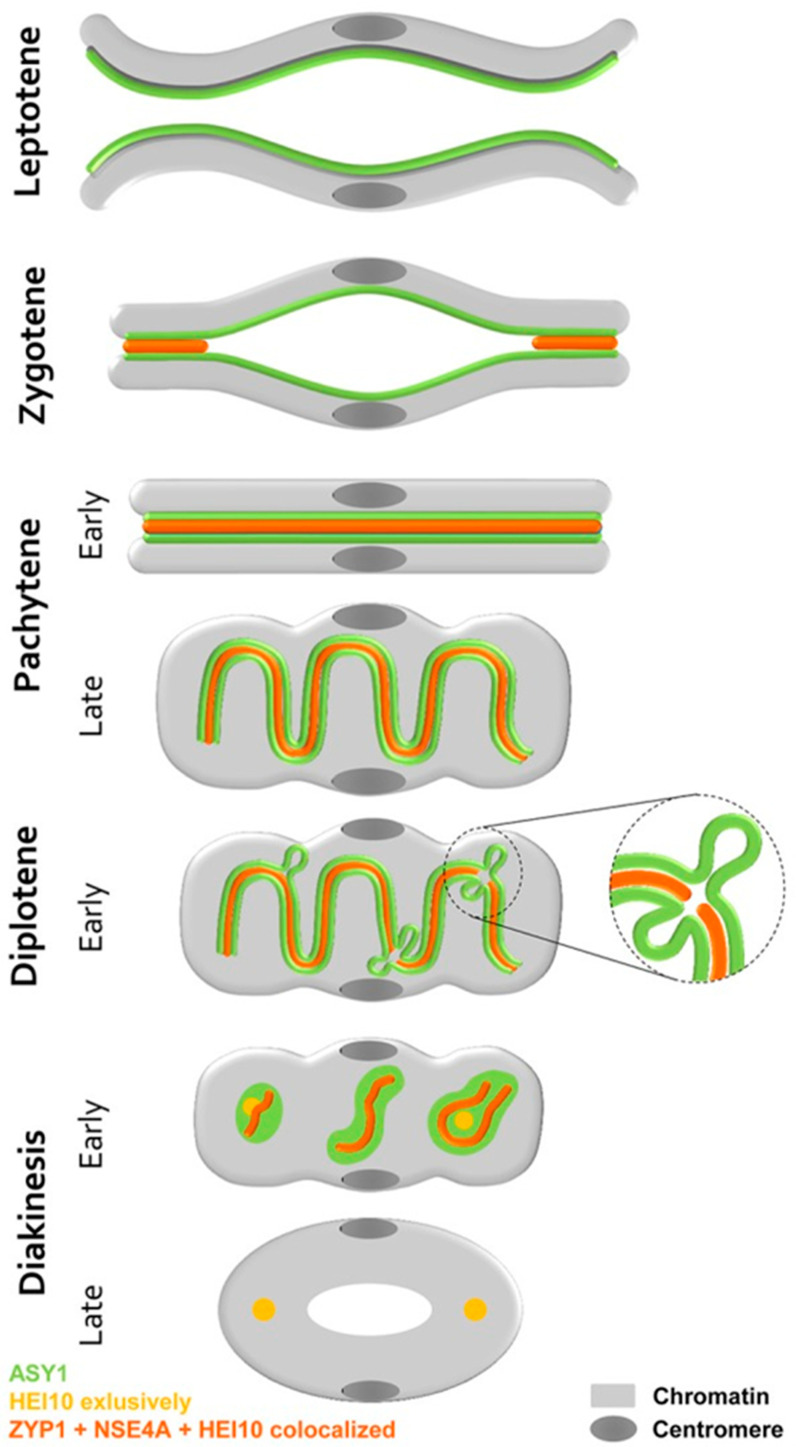
Schematic steps of homologous chromosome paring during meiotic prophase I including the proteins of the synaptonemal complex involved. Details are given by Hesse et al. (2019) [36]. Figure modified with permission.

**Figure 9 plants-11-01753-f009:**
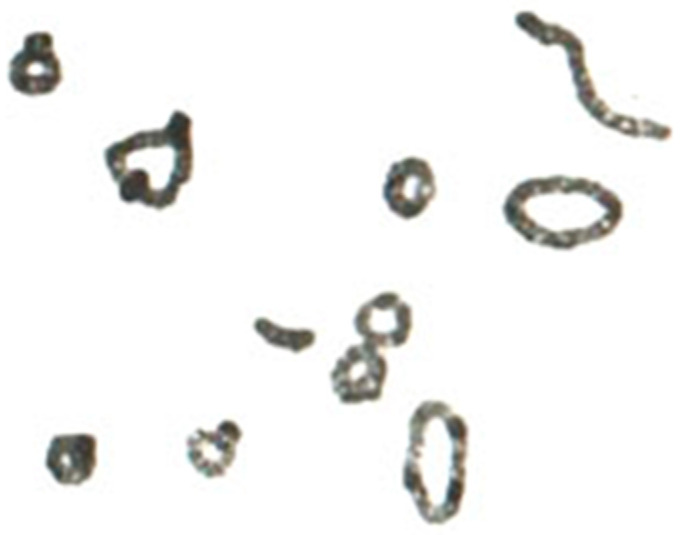
Multivalent chromosome associations in autotetraploid rye during MI of meiosis with ring and chain quadrivalents; chiasma frequency = 29 Xta/PMC.

**Figure 10 plants-11-01753-f010:**
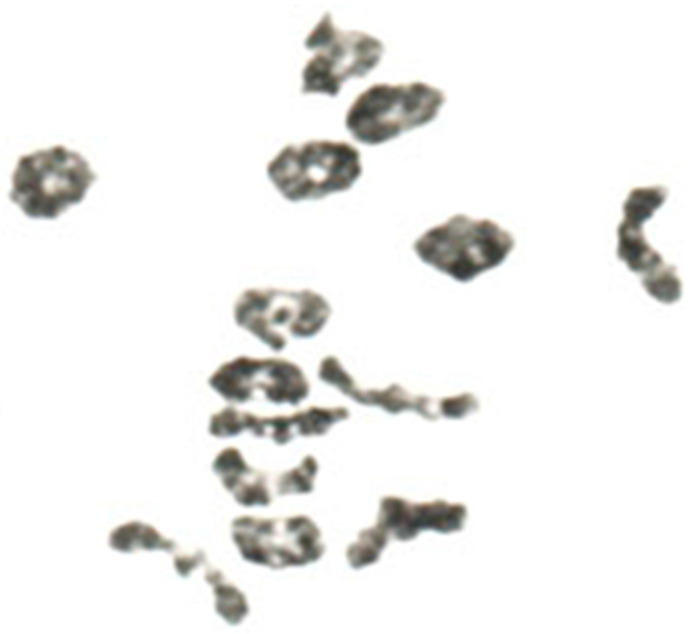
Diploid-like bivalent associations in autotetraploid rye during MI of meiosis after restructuring of the chromosome complement.

**Figure 11 plants-11-01753-f011:**
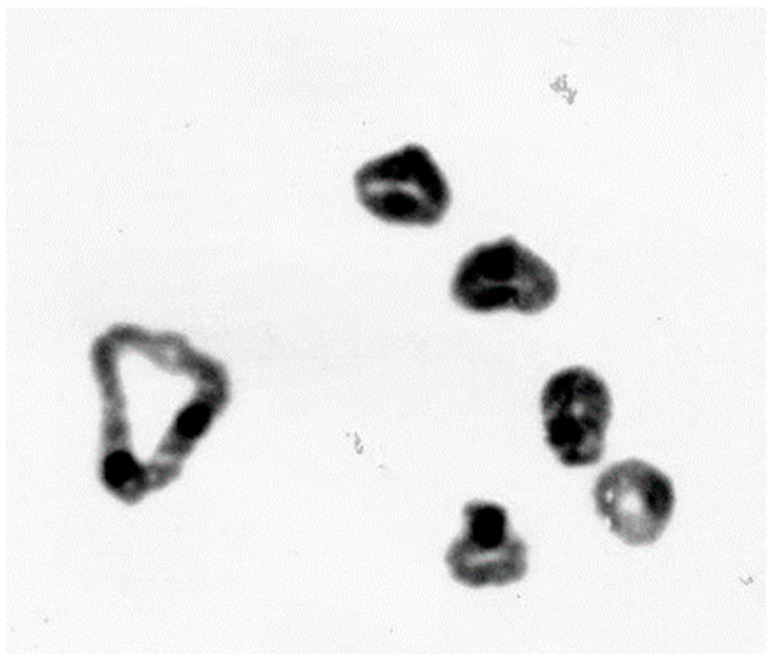
Metaphase I configuration in diploid rye, var. Petka, with 5-ring bivalents + 1-ring quadrivalent, caused by reciprocal translocation, after Feulgen staining.

**Figure 12 plants-11-01753-f012:**
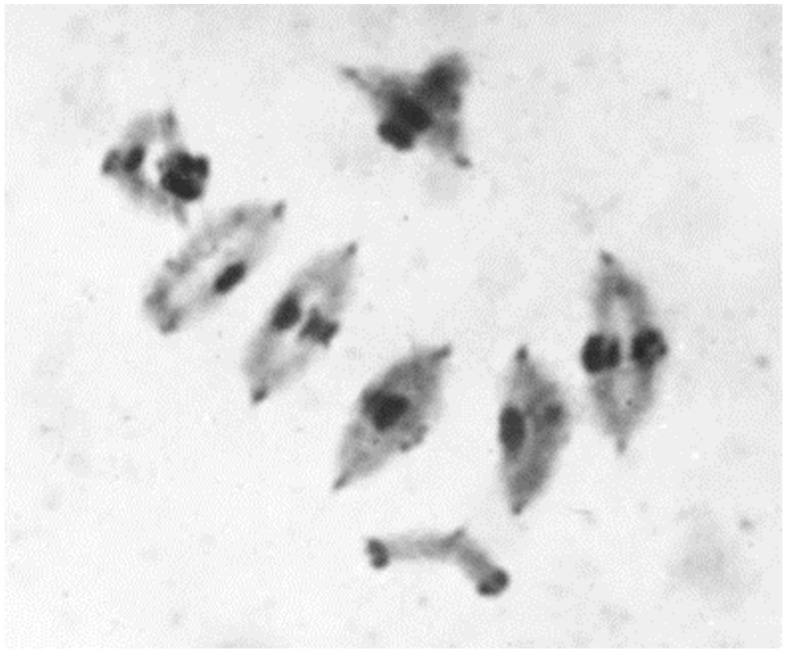
Meiotic spread of a trisomic rye plant (2*n* = 2x = 15) showing one additional univalent of chromosome 2R.

**Figure 13 plants-11-01753-f013:**
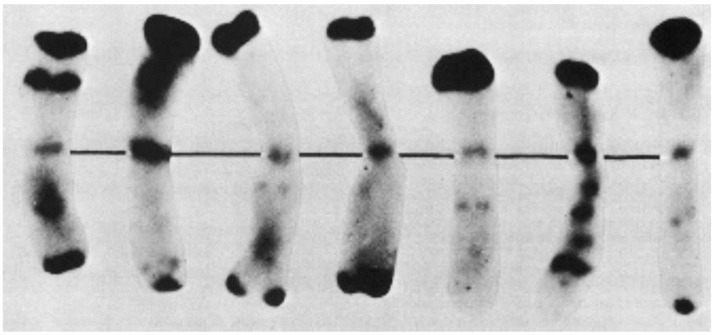
A first picture of the different rye chromosomes after Giemsa C-banding. The authors classified them according to the wheat–rye addition lines of Chinese Spring-Imperial from 1R (**left**) to 2R, 3R, 4R, 5R, 6R, and 7R (**right**). Modified after Gill and Kimber (1974), with permission.

**Figure 14 plants-11-01753-f014:**
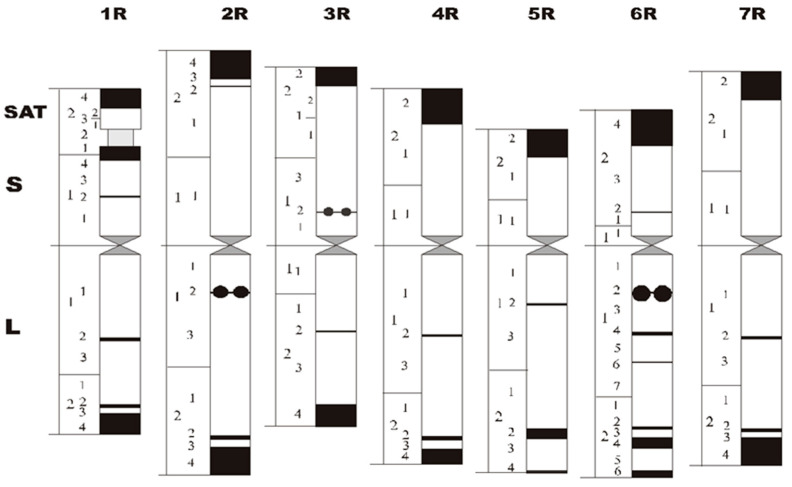
Schematic drawing of the complement of rye chromosomes with relative lengths of chromosomes and/or chromosome arms. Black blocks mark the C-bands, black dots mark the N-bands. Chromosomes are divided into regions/sub-regions and are described according to the system of human chromosomes.

**Figure 15 plants-11-01753-f015:**
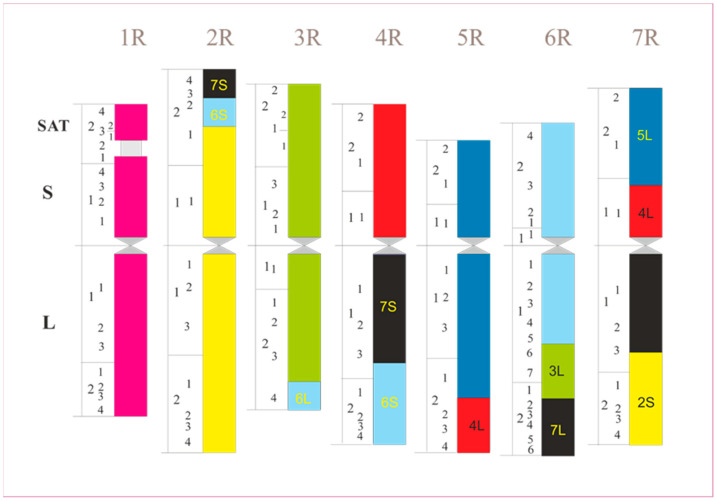
Schematic drawing of the complement of rye chromosomes with interchanged segments according to molecular analysis. Modified after Devos et al. (1993), with permission.

**Figure 16 plants-11-01753-f016:**
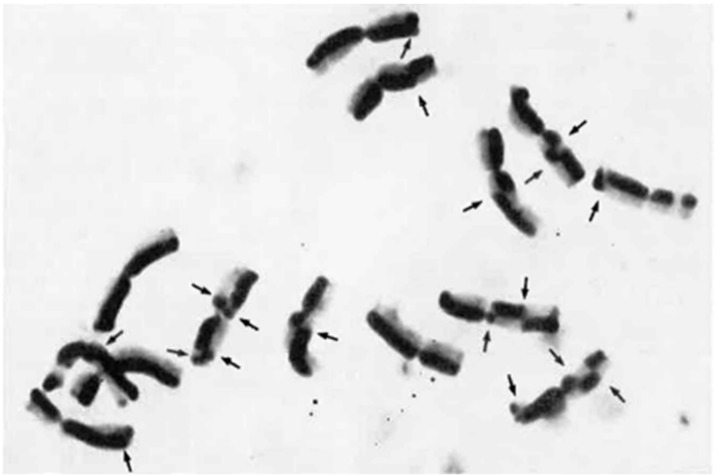
Mitotic metaphase chromosomes of *Secale cereale*, var. Karlshulder, with differential staining of the sister chromatids. Eighteen SCEs are recognizable (see arrows). Modified after Friebe (1978), with permission.

**Figure 17 plants-11-01753-f017:**
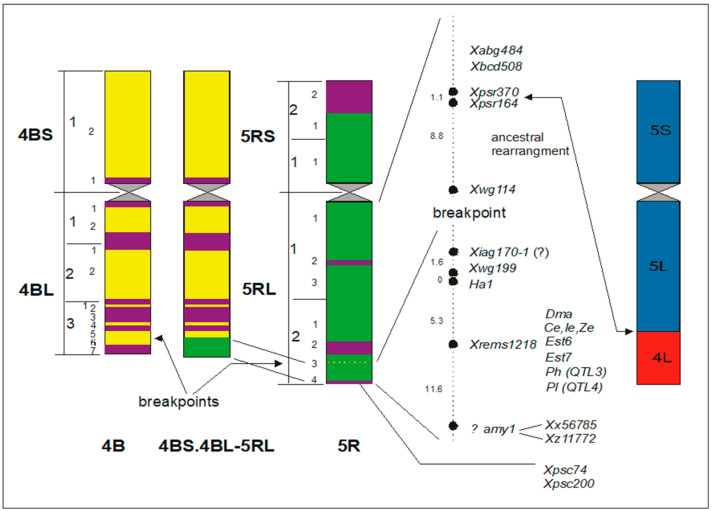
Schematic drawing of chromosomes in the wheat–rye translocation variety Viking after application of GISH staining. Considered chromosomes are 4B, 5R, and the 4BS.4BL-5RL translocation. Modified after Schlegel et al. 1993 [87].

**Figure 18 plants-11-01753-f018:**
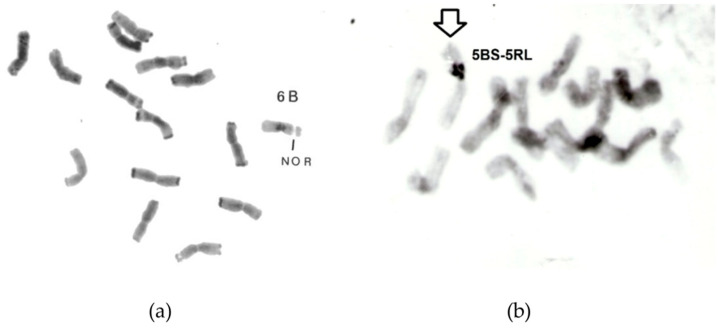
Rye–wheat whole chromosome addition line 6B, monosomic, 2*n* = 2x = 15, (**a**); heterozygote rye–wheat translocation line 5BS-5RL, 2*n* = 2x = 13rye + 1rye–wheat (**b**), after C- and/or N-banding.

**Figure 19 plants-11-01753-f019:**
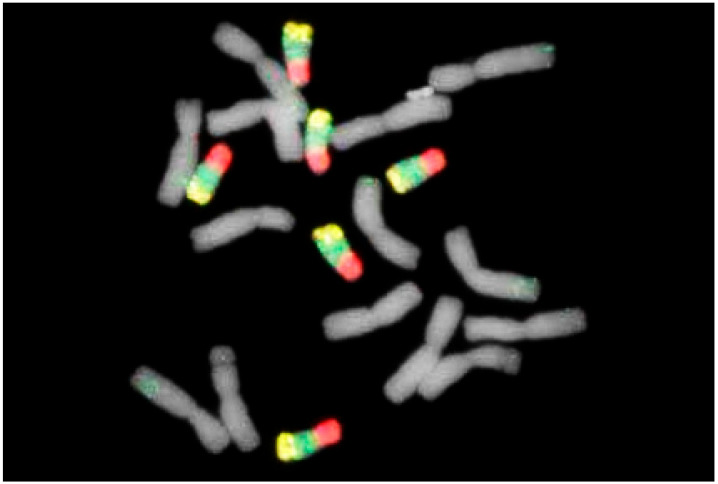
Rye plant (*Secale cereale*) with six B chromosomes after FISH with B-specific repeats of the D110ß0 family as probe. With permission of A. Marques and A. Houben, Gatersleben, Germany.

**Figure 20 plants-11-01753-f020:**
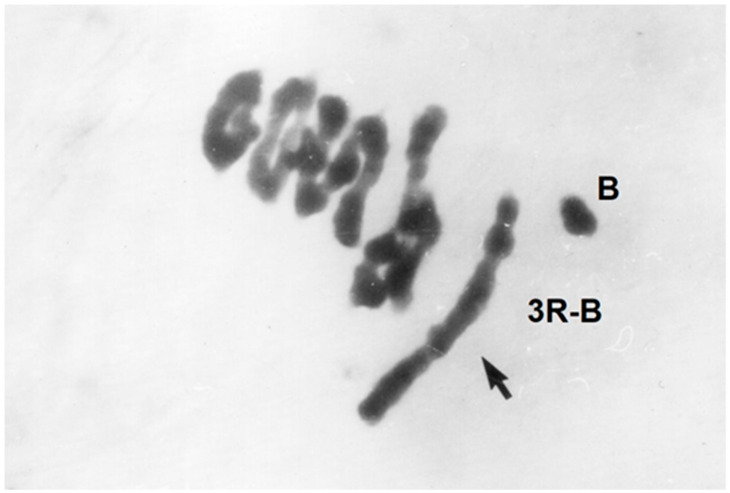
A–B translocation in Japanese rye JNK, 2*n* = 2x = 14A + 2B chromosomes during MI of meiosis with 3-ring bivalents + 1-rod bivalent + 1 chain quadrivalent + 1 heteromorphic trivalent (3R-3R-B) + 1 univalent (B), after Feulgen staining.

**Figure 21 plants-11-01753-f021:**
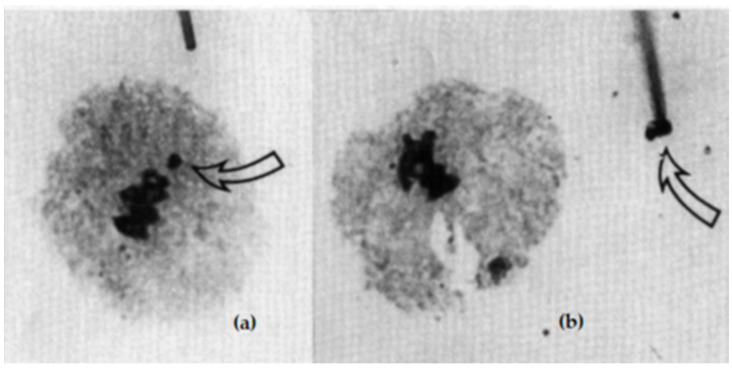
Pollen mother cell (MI) of a trisomic barley plant with 7 bivalents and 1 univalent, separately situated at cell equator (**a**). By micro-pipette, it can be extracted (see arrows) and used for further preparations (**b**). After Houben and Schlegel (1991), with permission.

**Figure 22 plants-11-01753-f022:**
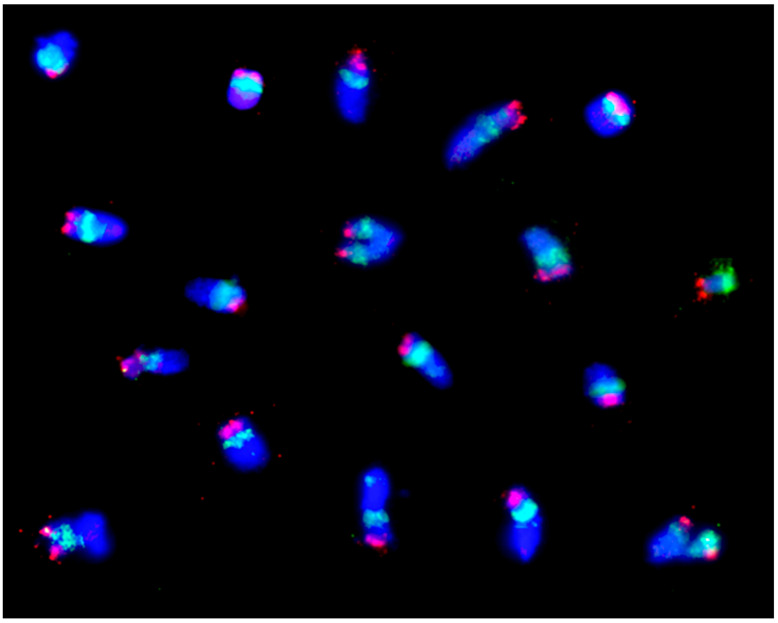
Microscopic spread of 1RS chromosome arms after flow sorting with a fluorescein isothiocyanate-labeled DNA probe. The nucleolus organizer region appears green. The red color marks the telomere. J. Dolezel, Olomouc, Czech Republic, with permission.

**Table 1 plants-11-01753-t001:** Homozygous reciprocal translocation lines of the rye variety Petkus defined by the chromosomes identified (modified after Sybenga et al. 1985 [44]).

Line No.	T273	T248	T305	T242	T240	T501	T282
Chromosomes/arm sinvolved	1RS–4RL, 1RL–5RS	1RS–5RL, 1RS–6RS	2RS–5RS, 2RL–5RS	2RL–6RL	3RS–5RS	3RS–5RL, 4RL–5RL	5RL–7RS

**Table 2 plants-11-01753-t002:** Chromosome nomenclature of rye compiled from different authors.

Chromosome							Reference
IV	V	I	III	VI	VII	II	[10]
2R	7R	3R	4R	5R	6R	1R	[62]
II	IV	III	I	VI	VII	V	[63]
1	2	3	4	5	6	7	[65]
I	III	II	V	IV	VI	VII	[66]
II	III	I	IV	VI	V	VII	[67]
b	g	c	a	e	d	f	[68]
A	B	F	G	D	E	C	[69]
2R	7R	3R	4R	5R	6R	1R	[70,71]
I	II	V	III	IV	VI	VII	[72]
2	3	5	1	4	6	7	[73]
I	IV	II	V	VI	III	VII	[74]

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
