# Peer review of "100 Years of Chromosome Research in Rye, Secale L."

_plants, 2022, doi:10.3390/plants11131753_

Round 1

Reviewer 1 Report

The manuscript is a good review of the methods used in cytogenetic studies in rye, and the major achievements made during the last 100 years.

The manuscript can be accepted in the present form after reviewing Figure number in line 403: Is it figure 11 or 14?

Author Response

The reviewer's comment on Fig. 11 was corrected in the manuscript to Figure 14.

Reviewer 2 Report

As the author states,  this chapter is a bit out of luck, having been beaten to the subject by recent earlier reviews, and ends up with a rather limited area: cytogenetics. The problem is, cytogenetics was left out because is out of fashion now. Consequently,  the relevance of this contribution is limited. However, historical accounts carry some value. In this sense I like this approach; I would just like a bit more of it, perhaps a somewhat different angle. Such as: explain why (technical limitations; different paradigms, etc) why early studies were mistaken. This can be done in more detail. Actually, given that cytogenetics is out of fashion and few people, if any, are familiar with terms and systems, perhaps it would make sense to explain things more clearly.

The chapter is what it is; the range and coverage are author's choice and preference, so perhaps it is none of my business. But I was asked an opinion.

My first impression is that this peace requires SERIOUS editorial work. I do not mean this as "editing of English language"; I mean it as general editing, for the way information is presented and organized. For example (and there are many similar): lines 454-459 are of of place and context; lines 468-9  a very odd paragraph break; odd phrasing line 386 "centromere sizes can vary 40-fold in genome size", sometimes things are left unexplained (lines 390-391 "phosphate level...."). There is just too much of it, as if written in a hurry. A professional editor will smooth it out.

I am sorry to say this but the manuscript strikes me as a bit self-serving. Yes, the author did make substantial contributions to rye cytogenetics, but just look at the numbers of referenced articles: 21 items for the author; no more than 2-3-4 for anyone else, including the biggest names in the field....

I did not think that I would ever hear "chiasma terminalization" mentioned seriously..... How odd....

Author Response

# The manuscrtipt was re-written considering the ideas of the reviewer

# Overlapping paragraphs were exchanged

# Several chapters were modified according to the  advises of the reviewer

Reviewer 3 Report

This manuscript reviewed the history of determining chromosome numbers in rye, ploidy, the meiotic behavior of rye chromosomes, chromosome staining, etc. This review provides some information about the study on rye chromosomes. However, this review was not prepared well and logical mess. For example, the section 4.4 is about chromosome staining, however, there is a paragraph is about the introgression of wheat chromosome into rye backgrounds. Therefore, this manuscript needs more careful reading and correction. There are some errors in this review. Major revisions are needed.

1.The description of the figure legends of Figure 7 is inconsistent with the figure. For example, "(b) 2 univalents and 5 ring and 1 rod bivalents". In fact, there are 6 ring bivalents in figure b. "(c) 6 univalents plus 2 ring and 2 rod bivalents". In fact, there are only two univalents in figure c.

2. The review on meiosis behavior of rye chromosomes is not complete. For example, Maestra et al. Chromosome Res., 2002, Corredor et al. Chromosome Res. 2007, Naranjo et al. Cytogenet. Genome Res., 2014. etc. were not been cited.

3. Line 403, “a specific reference karyogram was established (Figure 11)”. However, the Figure 11 is about the metaphase I configuration in diploid rye. It can not be used as reference karyogram.

4. The review on the advance of technology of identify rye chromosomes is not complete.

5. The review on the study about the structural variations of rye chromosomes is not complete.

6. The FISH karyotype of rye chromoosmes using tandem repeats as probes should be mentioned.

Author Response

# The manuscript was re-written considering the advises of the reviewer

# Fig. 7 was corrected; the confusion was caused by re-formating the manuscript

# For the chapter "meiosis" additional authors/data were considered

# The text link to the standard karyogram was corrected to Fig. 14

# Additonal data for advanced method of chromosome identifiation werde included

# Additonal data about structural variations of rye chromosomes were included

# FISH karyotyping with tandem repeats as probes was considered

Round 2

Reviewer 3 Report

This manuscript was revised well and it can be accepted for publication at present state.